# InfoDiffusion: Information Entropy Aware Diffusion Process for Non-Autoregressive Text Generation

**Renzhi Wang**[1,2]**, Jing Li**[3]**, Piji Li**[1,2*]

[1] College of Computer Science and Technology,
Nanjing University of Aeronautics and Astronautics, China
[2] MIIT Key Laboratory of Pattern Analysis and Machine Intelligence, Nanjing, China
[3] Department of Computing, The Hong Kong Polytechnic University, China
[1]{rzhwang,pjli}@nuaa.edu.cn
[3]jing-amelia.li@polyu.edu.hk

## Abstract

Diffusion models have garnered considerable interest in the field of text generation. Several studies have explored text diffusion models with different structures and applied them to various tasks, including named entity recognition and summarization. However, there exists a notable disparity between the "easy-first" text generation process of current diffusion models and the "keyword-first" natural text generation process of humans, which has received limited attention. To bridge this gap, we propose InfoDiffusion, a non-autoregressive text diffusion model. Our approach introduces a "keyinfo-first" generation strategy and incorporates a noise schedule based on the amount of text information. In addition, InfoDiffusion combines self-conditioning with a newly proposed partially noising model structure. Experimental results show that InfoDiffusion outperforms the baseline model in terms of generation quality and diversity, as well as exhibiting higher sampling efficiency.[1]

## 1 Introduction

Non-autoregressive (NAR) generation refers to a method of generating sequences where each element is generated independently, without relying on previously generated elements, allowing for faster parallel generation but potentially sacrificing the generation accuracy (Xiao et al., 2022). Recently, diffusion models have demonstrated powerful generative capabilities in image generation tasks, gradually becoming a new paradigm in generative models. The successful application of diffusion models to continuous data such as images and audio has motivated researchers to introduce them to discrete data like text. Previous researches have attempted to incorporate diffusion models into non-autoregressive text generation, designing different text diffusion model structures and applying them

to various text generation tasks, such as named entity recognition (Shen et al., 2023) and summarization (Zhang et al., 2023). However, these works have failed to recognize a fundamental difference between the process of generating text with diffusion models and the actual process of human text generation, which may be a reason why text diffusion models have consistently fallen short in terms of generation efficiency and quality.

Previous research has found that the text diffusion models seem to follow an "easy-first" principle (Emelianenko et al., 2019; He et al., 2022) in the decoding process. The "easy-first" principle means the model tends to generate tokens that are most frequently observed (and least surprising) in the training corpus, in order to achieve a higher likelihood. As the context becomes richer, more details are incorporated into the sequence. Figure 1 illustrates the decoding process of some existing text diffusion models, where it can be observed that the model tends to prioritize generating simple, high-frequency, and semantically poor words like "the" and "of" before generating less frequent but more informative and semantically rich words like "structure" and "remember". This differs from the actual order in which humans process or generate text. People tend to prioritize the core parts of a sentence or a paragraph, which contain crucial information (Grice, 1975). For example, when asked:"What are your upcoming plans?", you would answer:"I am going to finish a research paper." In this process, the words that come to your mind first are most likely "paper" or "finish", as they carry key information or have higher information entropy, rather than meaningless words like "a" or "the". It is difficult to imagine how someone whose first reaction is "the" would answer the question mentioned above. Similarly, this is also disastrous for a language model to which we expect to impart language abilities and even thoughts.

This inconsistent decoding order in text diffu-

---

*Corresponding author
[1]Code:https://github.com/rzhwang/InfoDiffusion

| D3PM | DiffusionBERT | DiffuSeq |
|---|---|---|

the man has also been arrested by the police .   today , he will be remembered for that mistake .   I want to become a good geologist .

the man has also **been** arrested by the police **.**   today **,** he will be remembered for **that** mistake **.**   I want to become **a** good geologist **.**

the man has also been arrested by **the** police .   **today** , he will **be** remembered for **that** mistake .   I want **to** become a good geologist .

the man **also** been arrested **by** the police .   today , he will be **remembered for** that mistake .   **I want** to become a **good** geologist .

**the** **man** **has** also been **arrested** by the **police** .   today , **he** will be remembered for that **mistake** .   I want to **become** a good **geologist** .

Figure 1: Inference process of three text diffusion models: illustrating the "easy-first" generation order. Each row represents one inference step.

sion models could lead to poor generation quality and low efficiency. On one hand, due to the fact that the core part of a sentence (key information) is accurately generated in the later half of the sampling process, the model lacks a comprehensive understanding of the overall semantic meaning of the sentence in the early stages, resulting in unsatisfactory generation quality. On the other hand, the lack of guidance from key information in the initial half of the sampling process leads to the generation of many meaningless or irrelevant words from the beginning. Due to the presence of these meaningless or even erroneous sampling steps, the efficiency of the model's sampling process is low.

To address the aforementioned issues, we propose a new non-autoregressive text generation model called **InfoDiffusion**. We devise a new noise schedule based on the information entropy of the text, enabling the model to aware the information carried by each word in a sentence. This guidance helps the model prioritize generating key information during the sampling process, thereby enhancing the quality and speed of sampling. Furthermore, we have integrated self-conditioning to further improve the generated output's quality and utilized "partially noising and conditional denoising" technique to realise sequence-to-sequence tasks.

In summary, our contributions are as follows:

- We propose a new non-autoregressive text generation model called InfoDiffusion, and enables the model to aware the information entropy contained in the text to prioritize generating key information during the sampling process.
- We combine self-conditioning and "partially noising and conditional denoising" to achieve high-quality sequence-to-sequence text generation.
- Experimental results demonstrate that InfoDiffusion, which follows a "keyinfo-first" generation order consistent with humans, achieves better generation quality and higher efficiency than baseline models across four text generation tasks.

## 2 Preliminaries

### 2.1 Diffusion Models

Diffusion models are a class of latent variable models characterized by a forward and a reverse Markov process (Sohl-Dickstein et al., 2015; Ho et al., 2020). In this framework, given a sample from the data distribution $x_0 \sim q(x_0)$, the forward process generates a sequence of latent variables $x_1, ..., x_T$ by sampling from:

$$q(x_t \mid x_{t-1}) = \mathcal{N}(x_t; \sqrt{1 - \beta_t} x_{t-1}, \beta_t \mathbf{I}) \quad (1)$$

where $\beta_t \in (0, 1)$ is a noise schedule controlling the step size of adding noise. Based on the reparameterization trick, arbitrary intermediate latent variable $x_t$ can be sampled in a closed form:

$$q(x_t \mid x_0) = \mathcal{N}(x_t; \sqrt{\bar{\alpha}_t} x_0, \sqrt{1 - \bar{\alpha}_t} \mathbf{I}) \quad (2)$$

where $\alpha_t = 1 - \beta_t$, $\bar{\alpha}_t = \coprod_{i=1}^{t} \alpha_i$. Following a predefined noise schedule, $\beta_t$ increases ($\alpha_t$ decreases) as the timestep grow and eventually corrupts $x_0$ into a random noise. If $\beta_t$ is small enough, the reverse process $q(x_{t-1}|x_t)$ is also a Gaussian, which is learned by a parameterized model:

$$p_\theta(x_{t-1} \mid x_t) = \mathcal{N}(x_{t-1}; \mu_\theta(x_t, t), \Sigma_\theta(x_t, t)) \quad (3)$$

where $\mu_\theta(x_t, t)$ and $\Sigma_\theta(x_t, t)$ can be implemented by a denoising networks $f_\theta(x_t, t)$ like U-Net or Transformer (Li et al., 2023). During inference, the reverse process begins with sampling noise from a Gaussian distribution $p(x_T) = \mathcal{N}(x_T; 0, \mathbf{I})$ and iteratively denoise it by $p_\theta(x_{t-1} \mid x_t)$ until obtaining $x_0$. The learning objective of diffusion models is derived from the variational lower bound of the negative likelihood of input $x_0$, denoted as:

$$\mathcal{L}_{vlb} = \mathbb{E}_q[D_{\mathrm{KL}}(q(x_T \mid x_0) \| p_\theta(x_T))]$$
$$+ \sum_{t=2}^{T} \mathbb{E}_q[D_{\mathrm{KL}}(q(x_{t-1} \mid x_t, x_0) \| p_\theta(x_{t-1} \mid x_t))]$$
$$- \mathbb{E}_q[\log p_\theta(x_0 \mid x_1)]$$

$$(4)$$

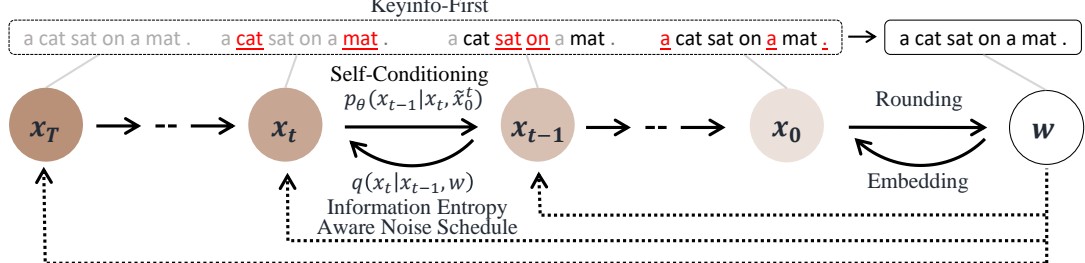

Figure 2: The overview of the proposed text diffusion model InfoDiffusion. Grey represents undecoded words, red underline indicates words decoded at the current time step, and black represents words decoded in previous time steps.

where $\mathbb{E}_q$ denotes the expectation over the joint distribution $q(x_{0:T})$. With additional condition on $x_0$, the posterior of the forward process $q(x_{t-1}|x_t, x_0)$ becomes tractable using Bayes theorem, then the simplified objective $L_{simple}$ can be expressed as:

$$\mathcal{L}_{simple} = \sum_{t=1}^{T} \mathbb{E}_q[\|\mu_t(x_t, x_0) - \mu_\theta(x_t, x_0)\|^2]$$

(5)

where $\mu_t$ is the mean of posterior $q(x_{t-1}|x_t, x_0)$. Through different parameterization strategies, the prediction objective can also be the noise (Ho et al., 2020) or original data $x_0$ (Li et al., 2022).

## 2.2 Continuous Text Diffusion Model

To adapt diffusion models for discrete text data, a straightforward approach is to employ word embedding, mapping discrete tokens to continuous word vector space before going through the continuous diffusion process. The continuous text diffusion model (Li et al., 2023), also known as the embedding diffusion model, introduces an embedding step $q_\phi(x_0|w) = \mathcal{N}(\text{EMB}(w), \sigma_0 \mathbf{I})$ in the forward process, where $\text{EMB}(w)$ represents a randomly initialized embedding function or obtained from a pre-trained model (such as BERT) that projects the discrete token $w$ into the continuous word vector space. For the backward process, the text diffusion model maps continuous word vectors back to their corresponding actual words through the word rounding module $p_\theta(w|x_0) = \coprod_{i=1}^{n} p_\theta(w_i|x_i)$. The inference process starts from random noise $x_T$ and follows the typical continuous diffusion process mentioned in Section 2.1 combined with word rounding to reconstruct the target word from the noise. To jointly learn the denoising network and the word embedding, the continuous text diffusion model extends the training objective in Equation 4

to a new end-to-end objective(Li et al., 2022):

$$\mathcal{L}_{vlb}^{e2e} = \mathbb{E}_q[\mathcal{L}_{vlb} + \log q_\phi(x_0|w) - \log p_\theta(w|x_0)]$$

(6)

which can be further simplified as:

$$\mathcal{L}_{simple}^{e2e} = \mathbb{E}_q[\mathcal{L}_{simple} + \|\text{EMB}(w) - \mu_\theta(x_1, x_0)\|^2 - \log p_\theta(w|x_0)]$$

(7)

## 3 InfoDiffusion

In this section, we introduce the detailed design of InfoDiffusion. The overall model architecture of InfoDiffusion is depicted in Figure 2. InfoDiffusion incorporates an Information Entropy Aware Noise Schedule, enabling the model to follow a "keyinfo-first" generation strategy, thereby achieving text generation that aligns with human-like processes. Additionally, InfoDiffusion combines self-conditioning and partially noising to achieve faster and superior text generation.

### 3.1 Information Entropy Aware Noise Schedule

In diffusion models, the noise schedule is a crucial component. The noise schedule directly determines how the original data is gradually perturbed during the forward process and how the model learns to recover the target data from the noise during the reverse process. This also leads to the significant influence of noise scheduling on the quality and diversity of generated samples. Previously, commonly used noise schedules, such as the linear schedule (Ho et al., 2020) and the cosine schedule (Nichol and Dhariwal, 2021), have shown promising results in image generation tasks. However, these schedules assume all tokens carry the same amount of information and does not consider the linguistic differences among the tokens in a sequence. This directly leads to a "shortcut" taken by

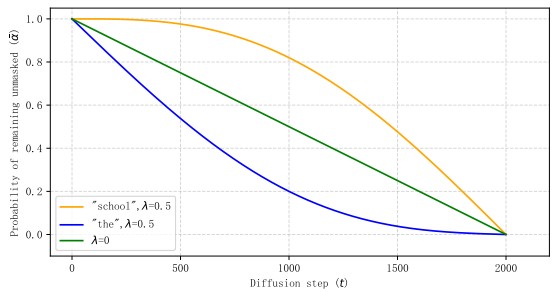

Figure 3: The noise schedule for each token in a sequence is determined based on the information entropy.

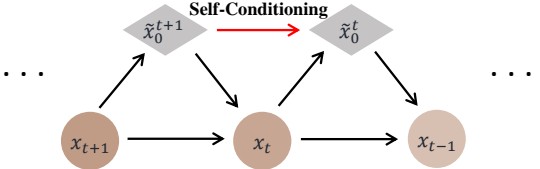

Figure 4: An illustration of reverse diffusion sampling steps with Self-Conditioning, sampling directly based on its previously generated samples.

the model: it tends to generate tokens that are most frequently appearing (and easiest) in the training corpus to achieve a higher likelihood. However, this generation order contradicts the behavioral pattern of humans, who tend to prioritize thinking about and generating the core parts of text, which contain higher information content. We refer to this human strategy of prioritizing the generation of high-information content parts in text as "key-information-first", abbreviated as "keyinfo-first".

In order to address the differences in the generation process mentioned above and enable the model to generate text more like human, we have designed a new noise schedule that can aware the informational entropy of words in a sentence. That is, at the initial stage of the forward process, low-information words are perturbed, and high-information words are perturbed at the final stage, thus guiding the model to prioritize generating key information during the reverse process.

Specifically, we first linearly interpolating the mutual information between the latent variables $x_t$ and the original data $x_0$ to 0, i.e. $I(x_t; x_0) \approx (1-\frac{t}{T})H(x_0)$, where $H$ denotes the entropy, which measures the amount of information of a random variable. In this case, the noise function in the classical diffusion model becomes $\beta = \frac{1}{T-t+1}$ (Equation 1) and $\bar{\alpha}_t = 1 - \frac{t}{T}$ (Equation 2) (Austin et al., 2021). Furthermore, to prioritize perturbing words with lower information before perturbing words with higher information, we design noise weights based on the information entropy of each word in a sentence $w$. The specific details are as follows:

$$\bar{\alpha}_t^i = 1 - \frac{t}{T} + \lambda(t)e(w^i) \in [0,1] \qquad (8)$$

$$\lambda(t) = \lambda \sin(\frac{t}{T}\pi) \qquad (9)$$

$$e(w^i) = \frac{H(w^i) - \bar{H}(w)}{max(H(w^j)) - min(H(w^j))} \qquad (10)$$

where $e(w^i)$ represents the normalized value of the information entropy of the $i$-th word in sentence $w$ and $\lambda(t)$ control the effect of the informativeness at time step $t$. To ensure that the latent variables $x_t$ retains all information at the beginning process ($t = 0$) and zero information at the end process ($t = T$), the noise schedule $\lambda(t)$ is designed to be sinusoidal, satisfying $\lambda(0) = \lambda(T) = 0$, following (He et al., 2022). The $\bar{H}(w)$ represents the mean entropy of sentence $w$ and $max(H(w^j))$ represents the maximum entropy in sentence $w$.

Figure 3 shows how $\bar{\alpha}_t$ progresses during the forward process. For example, consider the sentence "The bus passes by the school", the word "school" carries higher information content. Therefore, we encourage masking it at the later stage, allowing our model to learn to restore it in the early place.

It is worth noting that such a noise schedule does not alter the training objective since it does not modify the conditional probability function $q(x_{t-1}|x_t, x_0)$ in Equation 4.

## 3.2 Self-Conditioning

In the sampling process of classical diffusion models, at each time step $t$, the model generates the current prediction $\tilde{x}_0^t(x_t, t, \theta)$ through a denoising network $f_\theta(x_t, t)$. However, this denoising network only relies on the updated $x_t$ from the previous time step and discards the estimated result $\tilde{x}_0^{t+1}$, which means there is no connection between the predicted results of adjacent sampling steps. In the text generation process of the text diffusion model, this implies that the semantic information between the generated results of adjacent time steps is inconsistent and incoherent, inevitably leading to subpar text quality and low inference efficiency.

To address the issue of semantic incoherence mentioned above, inspired by (Chen et al., 2022), we employ the self-conditioning. As shown in Figure 4, this technique considers different denoising functions $\tilde{x}_0^t(x_t, \tilde{x}_0^{t+1}, t, \theta)$, utilizes the previously estimated samples as auxiliary inputs. Self-

conditioning refine the denoising function based on previous estimations instead of starting from scratch with new estimations. By doing so, direct connections and dependencies are established between the generation results of adjacent time steps, achieving semantic consistency and coherence. For more efficient model training, we adopt the same training strategy as Analog-Bit (Chen et al., 2022): with a 50% probability, we train $\tilde{x}_0^t(x_t, \tilde{x}_0^{t+1}, t, \theta)$ by setting the input $\tilde{x}_0^{t+1}$ to 0, which brings it back to the model without self-conditioning. Otherwise, we first estimate $\tilde{x}_0^t$ by $\tilde{x}_0^t(x_t, 0, t, \theta)$ and then use it for self-conditioning training. In the second case, we do not back-propagate through the first estimated $\tilde{x}_0^t$. Therefore, the increase in additional training time is less than 25%.

### 3.3 Partially Noising and Conditional Denoising

In the classical sequence-to-sequence task, given a source text $s = \{w_1^s, w_2^s, ..., w_n^s\}$ with n tokens, it generates target text sequence $y = \{w_1^y, w_2^y, ..., w_n^y\}$. A sequence generation model can achieve this by modeling the conditional probability: $p(y|s)$. To accomplish sequence-to-sequence text generation tasks, we employ the Partially Noising and Conditional Denoising (Gong et al., 2022). This technique adds noise only to the target text $y$ during forward process and applies denoising solely to $y$ during the denoising process.

Specifically, given a pair of text: the source text $w^s$ and the target text $w^y$, we first perform word embedding and concatenate the text pair as $\text{EMB}(w^s \odot y)$. Then, we obtain the initial state $x_0$ of the forward process through $q_\phi(x_0|w^x \odot y) = \mathcal{N}(\text{EMB}(w^s \odot y), \beta_0 \mathbf{I})$. To simplify the notation, we use $s_t$ and $y_t$ to represent the parts of $x_t$ belonging to $w^s$ and $w^y$ at diffusion time step $t$, following (Gong et al., 2022). In forward process, we only add noise to $y_t$ while keeping $s_t$ unchanged. In the reverse denoising process, $s_t$ is still kept unchanged and treated as the denoising condition, controlling and guiding the model to generate the desired text $y_t$ from the noise. The training objective at this point can be simplified as (Gong et al., 2022):

$$\mathcal{L}_{simple} = \sum_{t=2}^{T} [\|y_0 - f_\theta(x_t, t)\|^2] \\ + \|\text{EMB}(w^y) - f_\theta(x_1, 1)\|^2 \\ - \log p_\theta(w^s \odot y|x_0)$$

where $f_\theta$ is the denoising network.

## 4 Experimental Setup

### 4.1 Tasks and Datasets

Following (Gong et al., 2022), we conduct experiments on four typical and popular tasks: *Open domain dialogue*, *Question generation*, *Text simplification* and *Paraphrase*. **Open domain dialogue** requires models to generate informative and meaningful responses given a dialogue context. We employ the widely used Commonsense Conversation Dataset (Zhou et al., 2018), with over 3 million conversational pairs covering a wide range of everyday topics. **Question generation** aims to generate questions which can be answered by the given contents. We utilize the Quasar-T dataset (Dhingra et al., 2017), processed by (Gong et al., 2022), containing 119K training samples of document-question pairs. **Text simplification** aims to modify complex text into simplified sequences by simplifying grammar and word choice. We use the corpus constructed by (Jiang et al., 2020) consisting of 666K complex-simple sentences. **Paraphrase** involves rewriting sentence with the same semantic meaning but a different surface form. We adopt the widely used Quora Question Pairs (QQP), sourced from the community question-and-answer platform Quora, which consists of 147K positive pairs.

### 4.2 Baseline Methods

Following (Gong et al., 2022), we compare InfoDiffusion to four groups of baselines:

- **Encoder-decoder autoregressive model.** We choose two popular models: GRU (Chung et al., 2014) with attention and Transformer (Vaswani et al., 2017).

- **Fine-tuned large pre-trained language model.** We choose GPT-2 (Radford et al., 2019) and GPVAE (Du et al., 2022). GPT-2 is trained with language modeling and GPVAE augments T5 (Raffel et al., 2020) with VAE.

- **Non-autoregressive model.** we consider LevT (Cortes et al., 2015), a widely used, strong iterative NAR model. It adopts insertion and deletion to generate and refine sequences iteratively.

- **Text diffusion model.** We choose DiffuSeq (Gong et al., 2022). It is a recent text diffusion model, and the performance of other text diffusion models is similar to it.

We implement these models following their original papers.

Table 1: Evaluation results on four conditional text generation tasks. The best results are denoted by **bold** fonts, and the best results without pretrained language models are denoted by underline fonts.

| Dataset | Model | Quality | | | Diversity | | | Length |
|---|---|---|---|---|---|---|---|---|
| | | BLEU↑ | ROUGE-L↑ | BERTScore↑ | Dist-1↑ | Self-BLEU↓ | Diverse-4↑ | |
| Open Domain Dialogue | GRU-attention | 0.0068 | 0.1054 | 0.4128 | 0.8998 | 0.8008 | 0.1824 | 4.46 |
| | Transformer-base | **0.0189** | 0.1039 | 0.4781 | 0.7493 | 0.3698 | 0.6472 | 19.5 |
| | GPT2-base FT | 0.0108 | **0.1508** | 0.5279 | 0.9194 | 0.0182 | 0.9919 | 16.8 |
| | GPT2-large FT | 0.0125 | 0.1002 | **0.5293** | 0.9244 | 0.0213 | 0.9938 | 16.8 |
| | GPVAE-T5 | 0.0110 | 0.1009 | 0.4317 | 0.5625 | 0.3560 | 0.5551 | 20.1 |
| | NAR-LevT | 0.0138 | 0.0550 | 0.4760 | **0.9726** | 0.7103 | 0.1416 | 4.11 |
| | DiffuSeq | 0.0139 | 0.1056 | 0.5131 | 0.9467 | **0.0144** | **0.9971** | 13.6 |
| | InfoDiffusion | 0.0152 | 0.1272 | 0.5314 | 0.9497 | 0.0152 | 0.9810 | 15.3 |
| Question Generation | GRU-attention | 0.0651 | 0.2617 | 0.5222 | 0.7930 | 0.9999 | 0.3178 | 10.1 |
| | Transformer-base | 0.0364 | 0.1994 | 0.5334 | 0.8236 | 0.8767 | 0.4055 | 12.1 |
| | GPT2-base FT | 0.0741 | 0.2714 | 0.6052 | 0.9602 | 0.1403 | **0.9216** | 10.0 |
| | GPT2-large FT | 0.1110 | 0.3215 | **0.6346** | **0.9670** | 0.2910 | 0.8086 | 9.96 |
| | GPVAE-T5 | 0.1251 | 0.3390 | 0.6308 | 0.9381 | 0.3567 | 0.7286 | 11.4 |
| | NAR-LevT | 0.0930 | 0.2893 | 0.5491 | 0.8914 | 0.9830 | 0.4776 | 6.93 |
| | DiffuSeq | 0.1731 | 0.3665 | 0.6123 | 0.9056 | 0.2789 | 0.8103 | 11.5 |
| | InfoDiffusion | **0.1924** | **0.3892** | 0.6310 | 0.9142 | **0.2625** | 0.8021 | 12.7 |
| Text Simplification | GRU-attention | 0.3256 | 0.5602 | 0.7871 | 0.8883 | 0.9998 | 0.3313 | 18.9 |
| | Transformer-base | 0.2445 | 0.5058 | 0.7590 | 0.8886 | 0.8632 | 0.4028 | 18.5 |
| | GPT2-base FT | 0.3085 | 0.5461 | 0.8021 | 0.9439 | 0.5444 | 0.6047 | 16.1 |
| | GPT2-large FT | 0.2693 | 0.5111 | 0.7882 | **0.9464** | 0.6042 | 0.5876 | 15.4 |
| | GPVAE-T5 | 0.3392 | 0.5828 | 0.8166 | 0.9308 | 0.8147 | 0.4355 | 18.5 |
| | NAR-LevT | 0.2052 | 0.4402 | 0.7254 | 0.9715 | 0.9907 | 0.3271 | 8.31 |
| | DiffuSeq | 0.3622 | 0.5849 | 0.8126 | 0.9264 | 0.4642 | 0.6604 | 17.7 |
| | InfoDiffusion | **0.3941** | **0.5997** | **0.8437** | 0.9323 | **0.4515** | **0.6741** | 15.3 |
| Paraphrase | GRU-attention | 0.1894 | 0.5129 | 0.7763 | 0.9423 | 0.9958 | 0.3287 | 8.30 |
| | Transformer-base | 0.0580 | 0.2489 | 0.5392 | 0.7889 | 0.7717 | 0.4312 | 5.52 |
| | GPT2-base FT | 0.1980 | 0.5212 | 0.8246 | 0.9798 | 0.5480 | 0.6245 | 9.67 |
| | GPT2-large FT | 0.2059 | 0.5415 | 0.8363 | **0.9819** | 0.7325 | 0.5020 | 9.53 |
| | GPVAE-T5 | 0.2409 | 0.5886 | 0.8466 | 0.9688 | 0.5604 | 0.6169 | 9.60 |
| | NAR-LevT | 0.2268 | 0.5795 | 0.8344 | 0.9790 | 0.9995 | 0.3329 | 885 |
| | DiffuSeq | 0.2413 | 0.5880 | 0.8365 | 0.9807 | **0.2732** | 0.8641 | 11.2 |
| | InfoDiffusion | **0.2656** | **0.5928** | **0.8576** | 0.9815 | 0.2873 | **0.8972** | 11.4 |

## 4.3 Evaluation Metrics

When evaluating the generated sequences, both quality and diversity play vital roles. To assess quality, we employ BLEU (Papineni et al., 2002) and ROUGE (Lin, 2004) as standard metrics, measuring the overlapping n-grams between the generated and gold texts. However, since string matching alone may not suffice for open-ended generation, we also utilize BERTScore (Zhang et al., 2020) to evaluate the semantic similarity at the embedding level. Greater scores in BLEU, ROUGE, and BERTScore indicate superior performance in text generation. In terms of diversity, we consider evaluating distinct n-grams using Distinct (Li et al., 2016) and the ratio of distinct n-grams to total words using Diverse (Deshpande et al., 2019). Fur-

thermore, we incorporate self-BLEU (Zhu et al., 2018), a sentence-level metric that assesses overlapping n-grams among generated texts. A lower self-BLEU score and a higher diverse-4 value indicate a greater degree of diversity in the generated outputs. Following (Gong et al., 2022), we generate three samples per text condition to calculate the diversity metrics for each method.

## 4.4 Implementation Details

InfoDiffusion is built upon the 12 layers of Transformer with 12 attention heads and has approximately 91M parameters. The maximum sequence length is set to 128, with an embedding dimension of $d = 128$. We perform diffusion steps $T = 2,000$. To address out-of-vocabulary genera-

tion, we utilize Byte Pair Encoding (Sennrich et al., 2016) to construct the vocabulary. The accuracy metrics of InfoDiffusion are evaluated using MBR (Minimum Bayes Risk) with a candidate sample size of $|S| = 10$. The experiment is deployed on NVIDIA RTX 3090 Tensor Core GPUs, and we use 4 GPUs on training and single GPU on sampling.

## 5   Results and Analysis

### 5.1   Text Generation Evaluation

As shown in Table 1, we conclude that InfoDiffusion achieves comparable or even higher generation quality compared with strong baselines.

First, compared to encoder-decoder autoregressive models and Non-Autoregressive models, InfoDiffusion exhibits an absolute advantage in terms of quality and diversity. For instance, in question generation tasks, the quality metric BLEU has improved by more than threefold, while distinct has increased by +0.12. The improvement in diversity metrics is equally significant. For example, the value of diverse-4 increased from 0.64 to 0.98, representing an improvement of over 50%.

Second, compared to pre-trained models like GPT2, InfoDiffusion outperforms the base variant and performs comparably to the large variant, which has 8 times more parameters than InfoDiffusion. In terms of diversity, InfoDiffusion leads in seven out of the twelve comparative scenarios, indicating a slight advantage over pre-trained models in generating diverse texts.

Last, compared to the well-performing diffusion model DiffuSeq, InfoDiffusion demonstrates superior text generation quality across all datasets. All quality metrics show an improvement of +0.01 to +0.03. On the other hand, although the score of self-BLEU lags behind DiffuSeq in text simplification tasks, there is a slight improvement in text diversity across the remaining datasets.

### 5.2   Inference Efficiency Comparison

One of the major concerns of diffusion models is the efficiency of Inference. We compare our InfoDiffusion with DiffuSeq in terms of inference efficiency. We conduct experiments on Text Simplification and set the inference batch size to 50 and diffusion time step to 2000 for both models. The quality (i.e., BLEU) and diversity (i.e., div-4) curves during the model generation process are shown in the Figure 5. The quality and diversity of the text generated by DiffuSeq gradually im-

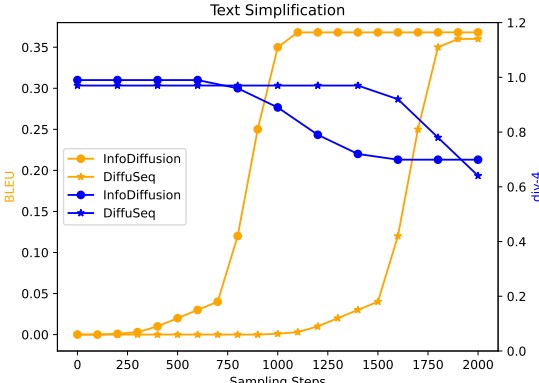

Figure 5: The curve of BLEU/div-4 score along with generation process.

Table 2: Ablation studies on QQP.

| Model | BLEU↑ | ROUGE-L↑ | BERTScore↑ | Dist-1↑ |
|---|---|---|---|---|
| InfoDiffusion | 0.2656 | 0.5928 | 0.8576 | 0.9815 |
| - Self-Conditioning | 0.2531 | 0.5884 | 0.8462 | 0.9816 |
| - Noise Schedule | 0.2480 | 0.5870 | 0.8413 | 0.9798 |

prove in the later stages of sampling (The decreasing trend in diversity metrics is due to the sampling process gradually generating the target text from noise and noise has a high level of diversity). But InfoDiffusion exhibits the opposite behavior, generating high-quality text in the early and middle stages of sampling. Approximately halfway through the sampling process, the quality of the text generated by InfoDiffusion surpasses the final results of DiffuSeq. This indicates that InfoDiffusion can converge to the target sentence more quickly and shorten the sampling time by half compared to DiffuSeq while maintaining almost the same generation performance.

### 5.3   Ablation Analysis

To demonstrate the effectiveness of the proposed techniques in InfoDiffusion, we conducted ablation studies on the QQP dataset. As shown in Table 2, when we removed the self-conditioning, the BLEU score decreased by 0.0126, while Dist-1 remained almost the same. Furthermore, when we additionally removed the proposed noise schedule from InfoDiffusion and used the $sqrt$ schedule proposed in DiffusionLM (Li et al., 2022) instead, the BLEU score dropped by 0.0051 and Dist-1 dropped by 0.0018. This indicates that the proposed noise schedule and self-conditioning contribute to improving the quality of generated text, while the impact of self-conditioning on the diversity of generated text is minimal.

Table 3: A sampling case from QQP dataset. We truncate the selected samples to the first 10 tokens and mark the generation process of each word with different colors.

| Diffusion Step $t$ | Generation Results of Intermediate Processes $\tilde{x}_0^t$ |
|---|---|
| Input Text | What should i do to be a great geologist? |
| $t = 100$ | athan backlash swiped i regentlated patrollingnine jennie ? chill [PAD] |
| $t = 130$ | athan backlash swiped i regentlated spotting geologist ? chilean [PAD] |
| $t = 200$ | clan patrice swiped i regent carmelgrowth geologist? [unused288] [PAD] |
| $t = 230$ | glancing patrice can i heringlated growth geologist ? navigable [PAD] |
| $t = 300$ | glance patrice can i moscowgrowth geologist ? corporal [PAD] [PAD] |
| $t = 340$ | [CLS] how can i 1859 a 1765 geologist? mcqueen [PAD] [PAD] [PAD] |
| $t = 400$ | [CLS] how can i [unused252] a sculpted geologist? [SEP] [PAD] [PAD] |
| $t = 490$ | [CLS] how can i 35th a nueva geologist? [SEP] [PAD] [PAD] |
| $t = 600$ | [CLS] how can i 35th a sculpted geologist? [SEP] [PAD] [PAD] |
| $t = 840$ | [CLS] how can i become a good geologist? [SEP] |
| $t = 950$ | [CLS] how can i become a good geologist? [SEP] |
| $t = 1000$ | [CLS] how can i become a good geologist? [SEP] |
| $t = 1600$ | [CLS] how can i become a good geologist? [SEP] |
| $t = 2000$ | [CLS] how can i become a good geologist ? [SEP] |

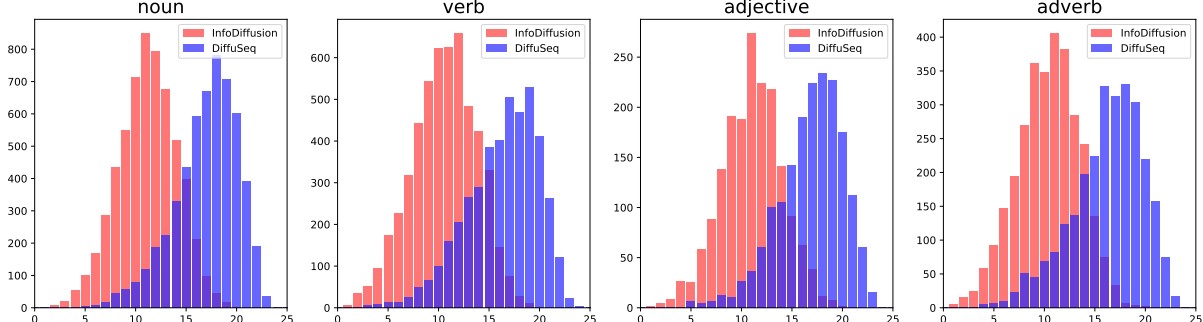

Figure 6: Comparison of distributions of different types of words relative generation order. The x-axis represents the diffusion step $t$, while the y-axis represents the number of words of a certain type that are first decoded in that diffusion step.

## 5.4 Case Study

We select an illustrative cases and investigate the generation process of InfoDiffusion. There are more cases in the Appendix C. As shown in Table 3, the generation process reveals that the InfoDiffusion model follows the "keyinfo-first" generation order: it prioritizes generating nouns with higher information content, such as "i" and "geologist", and then sequentially generates words with lower information content, such as "can", "how", "become", and "good" to complement the sentence.

In order to illustrate more clearly the model's preference for generating key information, we selected four categories of words that generally have key information or higher information content: nouns, verbs, adverbs, and adjectives (Clark and Weir, 2002; Emelianenko et al., 2019). We compared the decoding order of these words in InfoDiffusion and DiffuSeq during the decoding process. As shown in Figure 6, it is evident that InfoDiffusion decodes these high-information words much more earlier compared to DiffuSeq.

## 6 Conclusion

This paper, we propose InfoDiffusion, a novel non-autoregressive text diffusion model. By designing an Information Entropy Aware Noise Schedule, we enable the diffusion model to follow a "keyinfo-first" text generation process that is more aligned with human text generation, thereby achieving improved effectiveness and efficiency in text generation. Experimental results on four benchmark datasets confirm the effectiveness of InfoDiffusion. This study is the first research on the decoding order of diffusion models and the first attempt to alter the decoding order of diffusion text models. Future work could explore the use of the proposed noise schedule to replace the existing noise in related tasks based on diffusion models, in order to further enhance the model's performance.

## Limitations

Despite the strong performance of InfoDiffusion, it still has the following limitations. First, due to the strong preference of language for simple words, simple words may still appear early in the decoding process. Second, our evaluation relied solely on automatic metrics like BLEU, without assessing potential issues like hallucinations in the generated text. Future work could utilize both automatic metrics and human evaluation to comprehensively assess text quality across dimensions including grammar, semantics, and more. This multifaceted approach will facilitate the generation of truthful, logical, and reliable text.

## Ethics Statement

The research presented in this paper on the diffusion text model adheres to ethical guidelines and principles. We have prioritized privacy, mitigated biases, ensured transparency, and promoted responsible use. Our commitment to accountability, responsible governance, and continuous ethical assessment underscores our dedication to upholding the highest standards of integrity in the development and deployment of the diffusion text model.

## Acknowledgements

This research is supported by the National Natural Science Foundation of China (No.62106105), the CCF-Baidu Open Fund (No.CCF-Baidu202307), the CCF-Tencent Open Research Fund (No.RAGR20220122), the CCF-Zhipu AI Large Model Fund (No.CCF-Zhipu202315), the Fundamental Research Funds for the Central Universities (No.NJ2023032), the Scientific Research Starting Foundation of Nanjing University of Aeronautics and Astronautics (No.YQR21022), and the High Performance Computing Platform of Nanjing University of Aeronautics and Astronautics.

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

## A  Related Works

### A.1  Text Diffusion Models

Adapting diffusion models to non-autoregressive (NAR) text generation poses a challenge due to the discrete nature of text. Discrete tokens cannot be directly corrupted by continuous noise, necessitating the redesign of typical diffusion models for text data. In this section, we focus on diffusion models customized for text, which can either perform diffusion in discrete space or incorporate an additional step of mapping discrete tokens to latent space of token embeddings before applying continuous diffusion.

**Discrete Text Diffusion Models.** These text diffusion models extend diffusion models to discrete state spaces by corrupting and refining the sentences at the token level. D3PM (Austin et al., 2021) employs Markov transition matrices instead of Gaussian noise to diffuse real-world distributions. DiffusER (Reid et al., 2022) generates a sequence of edits that effectively transforms a random noise into output. DiffusionBERT (He et al., 2022) combines diffusion models with pre-trained Language Models to enhance their performance. Diffusion-NAT (Zhou et al., 2023) proposes an iterative self-prompting strategy for denoising process.

**Continues Text Diffusion Models.** Continuous text diffusion models introduce an additional step where discrete tokens are mapped to the latent space of token embeddings, followed by the adoption of continuous diffusion. Diffusion-LM (Li et al., 2022) is the first to proposes constructing diffusion models on continuous word embedding space. DiffuSeq (Gong et al., 2022) focuses on sequence-to-sequence generation using encoder-only Transformers and utilizes partial noising to define the diffusion process and learn the denoising function. SeqDiffSeq (Yuan et al., 2022) proposes an encoder-decoder diffusion model architecture for conditional generation and uses adaptive noise schedule technique to improve generation quality. DiNoiSer (Ye et al., 2023) proposes an adaptive method to determine the range of noise scales sampled for counter-discreteness training allowing the model to leverage amplified noise scales from the source conditions during inference. Masked Diffuse LM (Chen et al., 2023) follows easy first generation and designs a soft masking strategy based on tf-idf. Additionally, DiffuSum (Zhang et al., 2023) applies diffusion models to enhance extractive summarization.

Table 4: Examples of sampling process of InfoDiffusion.

| Input Text | How do I read and find my YouTube comments? |
|---|---|
| $t = 400$ | [CLS] how can i aground dismissing youtube johnstone? hammond [PAD] [PAD] |
| $t = 800$ | [CLS] how can i increase [unused336] youtube 1855? [SEP] |
| $t = 1200$ | [CLS] how can i read my youtube comments? [SEP] |
| $t = 1600$ | [CLS] how can i read my youtube comments? [SEP] |
| $t = 2000$ | [CLS] how can i read my youtube comments? [SEP] |
| Input Text | How do i make friends? |
| $t = 400$ | [CLS] how bassett iseibner anthropologist casino? [SEP] [PAD] |
| $t = 800$ | [CLS] how do i the my friends iidae? [SEP][PAD][PAD] |
| $t = 1200$ | [CLS] how do i make my friends? [SEP] |
| $t = 1600$ | [CLS] how do i make my friends? [SEP] |
| $t = 2000$ | [CLS] how do i make my friends? [SEP] |
| Input Text | How i can speak english fluently? |
| $t = 400$ | [CLS] how can i campos susceptible pauline and brien gunfire? [SEP] sarawak forts [PAD] [PAD] |
| $t = 800$ | [CLS] how can i campos robyn english outlaws scrambled forts? [SEP] |
| $t = 1200$ | [CLS] how can i speak fluent english and get tammy? [SEP] |
| $t = 1600$ | [CLS] how can i speak fluent english and get confident? [SEP] |
| $t = 2000$ | [CLS] how can i speak fluent english and get confident? [SEP] |

Table 5: Examples of sampling process of DiffuSeq.

| Input Text | How do I read and find my YouTube comments? |
|---|---|
| $t = 400$ | [CLS] docks i doo blessed [unused305] snoop sentencing creators [PAD] [PAD] nall [PAD]wangains [unused781] |
| $t = 800$ | [CLS] how nods i belgrade 139 ratios 2 ? [SEP] [PAD] [PAD] [PAD] [PAD] heresy [PAD] |
| $t = 1200$ | [CLS] how gymnasium i laguna [unused730] youtube lobbied? [SEP]] |
| $t = 1600$ | [CLS] how do i 275 my [unused730] comments? [SEP] |
| $t = 2000$ | [CLS] how do i read my youtube comments? [SEP] |
| Input Text | How do i make friends? |
| $t = 400$ | stamp trinidad i glint labyrinth [unused347] admiral [PAD] [PAD] [unused481] joo rochester governors examines [PAD] [PAD] |
| $t = 800$ | [CLS] blackout charlton i ames labyrinth? [SEP] [PAD] |
| $t = 1200$ | [CLS] howł i engraving guinness? [SEP] |
| $t = 1600$ | [CLS] how bonnet i engraving guinness? [SEP] |
| $t = 2000$ | [CLS] how do i make my friends? [SEP] |
| Input Text | How i can speak english fluently? |
| $t = 400$ | [CLS] tehran dispatched northernmost oswald lansing jennie fivb whistling routing [PAD]ikh [PAD] [PAD] sheds |
| $t = 800$ | [CLS] ton i efficacy battled city and? [SEP] [PAD] [PAD]ikh [PAD] [PAD] |
| $t = 1200$ | [CLS] how [unused293] i discovers dario madden english? [SEP] |
| $t = 1600$ | [CLS] how can i oswald fluent [unused490] english? [SEP] |
| $t = 2000$ | [CLS] how can i speak fluent like english? [SEP] |

## A.2 Noise schedule

Various noise schedules have been proposed to control the frequency of different input data in the denoising network. Existing methods either adopt popular noise schedules from vision tasks or design new schedules specifically tailored to text data.

**Linear Schedule.** The linear schedule (Ho et al., 2020) involves varying $\beta_t$ from $10^{-4}$ to 0.02 in a linear manner. This schedule is designed to maintain a relatively low noise scale at the start.

**Cosine Schedule.** This schedule defines $\bar{\alpha}_t = \frac{f(t)}{f(0)}$, where $f(t) = cos(\frac{t/T+s}{1+s} \cdot \frac{\pi}{2})^2$ to slows down the growing speed of the noise scale.

**Mutual Information Schedule.** D3PM (Austin et al., 2021) designs the mutual information schedule for discrete diffusion models with absorbing states. This schedule reduces to $\beta_t = \frac{1}{T-t+1}$, as same as in (Sohl-Dickstein et al., 2015).

**Sqrt Schedule.** Diffusion-LM (Li et al., 2022) presents the $sqrt$ schedule, which is defined as $\bar{\alpha}_t = 1 - \sqrt{t/T + s}$. This noise schedule is also adopted by DiffuSeq (Gong et al., 2022).

**Adaptive Schedule.** SeqDiffSeq (Yuan et al., 2022) proposes an approach to increase the difficulty of predicting $x_0$ in correlation with the timestep. This schedule involves learning the relationship between the noise scale and the loss using

Table 6: Examples of sampling process on LM1B.

| Model | Sampling Process |
|---|---|
| DiffusionBERT | [MASK] [MASK] [MASK] [MASK] [MASK] [MASK] [MASK] [MASK] [MASK] [MASK]
[MASK] , [MASK] [MASK] [MASK] [MASK] [MASK] that [MASK] .
today , [MASK] will be [MASK] [MASK] that [MASK] .
today , [MASK] will be remembered for that mistake .
today , he will be remembered for that mistake . |
| InfoDiffusion-discrete | [MASK] [MASK] [MASK] [MASK] [MASK] [MASK] [MASK] [MASK]
I [MASK] [MASK] [MASK] [MASK] [MASK] mistakes [MASK]
I [MASK] [MASK] [MASK] their [MASK] mistakes .
I make up for their [MASK] mistakes .
I make up for their recent mistakes . |

Table 7: The comparison for different models.

| Model | Parameters | Learning Paradigm | Diversity Source |
|---|---|---|---|
| GRU-attention | 65M | encoder-decoder | - |
| Transformer-base | 80M | encoder-decoder | Temperature |
| GPT2-base FT | 117M | pretrain-finetune | Hybrid strategy |
| GPT2-large FT | 117M | pretrain-finetune | Hybrid strategy |
| GPVAE-T5 | 117M | pretrain-finetune | Gaussian sampling |
| NAR-LevT | 117M | pretrain+VAE | - |
| DiffuSeq | 91M | non-autoregressive | Gaussian sampling |
| InfoDiffusion | 92M | non-autoregressive | Gaussian sampling |

Table 8: Experimental results on LM1B.

| Model | PPL↓ | BLEU↑ | Self-BLEU↓ |
|---|---|---|---|
| D3PM (Austin et al., 2021) | 77.50 | 0.4241 | 0.2288 |
| Diffusion-LM (Li et al., 2022) | 118.62 | 0.3553 | 0.2668 |
| DiffusionBERT (He et al., 2022) | **63.87** | 0.4358 | 0.2151 |
| InfoDiffusion-discrete | 68.46 | **0.4529** | **0.2084** |

Table 9: Experimental results on Quasar-T and QQP.

| Dataset | Model | BLEU↑ | Rouge-L↑ |
|---|---|---|---|
| Quasar-T | DiffuSeq (Gong et al., 2022) | 0.1731 | 0.3665 |
| | DiffusionBERT (He et al., 2022) | 0.0971 | 0.3420 |
| | InfoDiffusion | **0.1924** | **0.3892** |
| QQP | DiffuSeq (Gong et al., 2022) | 0.2431 | 0.5880 |
| | DiffusionBERT (He et al., 2022) | 0.2420 | 0.5845 |
| | InfoDiffusion | **0.2656** | **0.5928** |

Table 10: Experimental results on PersonaChat, XSUM and SQuAD.

| Model | PersonaChat | | XSUM | SQuAD |
|---|---|---|---|---|
| | BLEU-1↑ | BLEU-2↑ | Rouge-L↓ | Rouge-L↓ |
| DiffuSeq | 37.79 | 32.50 | 20.29 | 29.29 |
| InfoDiffusion | **40.13** | **36.47** | **24.96** | **35.70** |

an existing schedule (such as the $sqrt$ schedule).

## B Baselines Settings

We follow the same baseline settings as (Gong et al., 2022) and the results are also collected from their work. The settings are listed in Table 7. The GRU-attention encoder-decoder model lacks diversity search algorithms, resulting in limited sentence-level diversity. In the case of NAR-LevT, we adhere to the original paper's methodology by setting the maximum iteration to 9 and applying the specified termination condition. As for GPVAE-T5, we assign a scalar value of 2 to all tasks.

## C Inference Cases

In this section, we provide more sampling examples of InfoDiffusion and DiffuSeq to illustrate different sampling processes of text diffusion models.

From the Table 4 and Table 5, we can visually observe that InfoDiffusion has the ability to prioritize the generation of high-information words and converge to the target sentence more quickly.

## D Additional Experiments

Following the reviewers' advice and (Ni et al., 2023; Ni and Li, 2023; Yin et al., 2023), in this section, we conduct some additional experiments to further demonstrate the effectiveness and generalization of the proposed method.

We validate the efficacy of the proposed method on discrete text diffusion models. To implement our method on discrete diffusion models, we adopt some settings from DiffusionBERT and adjust the noise schedule and the decoding schemes. On one hand, we train our model on LM1B (Chelba et al., 2014) which was used by DiffusionBERT, and the experimental results are shown in Table 6 and Table 8. On the other hand, we also compare the performance of DiffusionBERT on Quasar-T (Dhingra et al., 2017) and QQP datasets used in our paper, and the results are shown in Table 9. The experimental results show that the method we proposed is also effective in the discrete domain. The

model can follow the "keyinfo-first" generation order while maintaining high quality, showing the general applicability of our approach.

Meanwhile, we conduct experiments on another 3 datasets: PersonaChat (Zhang et al., 2018), XSUM (Narayan et al., 2018) and SQuAD (Rajpurkar et al., 2016). PersonaChat is a dataset for dialogue generation, with the goal of predicting responses according to the dialog history. XSUM is a dataset for summarization, with the goal of summarizing the document into a sentence. SQuAD is a dataset for question generation, with the goal of generating questions based on given passages and answers. The results in Table 10 demonstrate that InfoDiffusion still achieves strong performance across these datasets.