# OpenReview forum: "InfoDiffusion: Information Entropy Aware Diffusion Process for Non-Autoregressive Text Generation"
_EMNLP/2023/Conference — EMNLP 2023 Findings_

### Official Review · Reviewer_SUds · 2023-08-03

**Soundness:** 3

**Excitement:**

3: Ambivalent: It has merits (e.g., it reports state-of-the-art results, the idea is nice), but there are key weaknesses (e.g., it describes incremental work), and it can significantly benefit from another round of revision. However, I won't object to accepting it if my co-reviewers champion it.

**Missing References:**

other info-based diffusion language modeling methods.

**Paper Topic And Main Contributions:**

This paper supposes the easy word first generation process affects the generation quality, so they:
1. proposes an info-rich keywords first generation schedule;
2.. combine some superior tricks to boost the performance;
3. The experiment results are good.

**Questions For The Authors:**

I am not so clear about the difference between your work and other info-based diffusion language modeling methods.

**Reasons To Accept:**

1. the idea is technique sound.
2. the whole logic is easy to follow, although the abstract is not so clear.

**Reasons To Reject:**

1. The abstract is not well organized, which make me confused, e.g., what's the notable disparity?
2. What are the differences between your paper and some others that appeared before you by using info entropy to guide the diffusion process?
3. In principle, the keyword first generation sounds reasonable, but if the first keywords are not right, the whole generation will become serious, how to avoid this phenomenon?
4. The key contribution is the info-rich word generation, what the advantages and disadvantages over the efficient methods, like dictionary-based methods or tf-idf-based methods.

**Reproducibility:**

4: Could mostly reproduce the results, but there may be some variation because of sample variance or minor variations in their interpretation of the protocol or method.

**Reviewer Confidence:**

3: Pretty sure, but there's a chance I missed something. Although I have a good feel for this area in general, I did not carefully check the paper's details, e.g., the math, experimental design, or novelty.

---

> ### Author Rebuttal · Authors · 2023-08-27
>
> Thank you for taking the time to review my paper and provide thoughtful feedback. I appreciate you recognizing the technical soundness and logical flow of the approach. Let us address your questions one by one.
>
> > Q1: Abstract is not well organized.
>
> A1：We will revise the abstract to clearly point out that the notable disparity refers to diffusion models prioritizing the generation of meaningless words, while human linguistic habits prioritize thinking about keywords.
>
> > Q2: Differences between your paper and some others by using info entropy to guide the diffusion process？
>
> A2：Our method places more emphasis on the "keyinfo-first" strategy of prioritizing the generation of keywords. To measure whether a word is a keyword, we assume that words with higher information entropy are more critical and important to a sentence. Therefore, we use information entropy as the control. Measuring key information through methods other than info entropy is also feasible. Meanwhile, the models proposed in other works, such as Masked-Diffuse LM[1] and DiffusionBERT[2], still follow an easy-first generation order. On the other hand, the "keyinfo-first" strategy we propose can also be combined with other strategies using info entropy to achieve better results. For example, we combined DiffusionBERT and implemented a discrete version of InfoDiffusion to further demonstrate the effectiveness and generalizability of the proposed method. The experimental results are shown in the table below.
>
> | Method                 | PPL$\downarrow$ | BLEU$\uparrow$ | Self-BLEU$\downarrow$ |
> | ---------------------- | --------------- | -------------- | --------------------- |
> | D3PM[3]                | 77.50           | 0.4241         | 0.2288                |
> | Diffusion-LM[4]        | 118.62          | 0.3553         | 0.2668                |
> | DiffusionBERT          | **63.78**       | 0.4358         | 0.2151                |
> | InfoDiffusion-discrete | 68.46           | **0.4529**     | **0.2084**            |
>
> In addition to the differences mentioned above, InfoDiffusion also outperforms other models using information entropy in terms of performance. For example, we compare the performance of InfoDiffusion with DiffusionBERT on the Quasar-T and QQP datasets as follows.
>
> | Dataset  | Method        | BLEU$\uparrow$ | Rouge-L$\uparrow$ |
> | -------- | ------------- | -------------- | ----------------- |
> | Quasar-T | DiffusionBERT | 0.0971         | 0.3420            |
> |          | InfoDiffusion | **0.1924**     | **0.3892**        |
> | QQP      | DiffusionBERT | 0.2420         | 0.5845            |
> |          | InfoDiffusion | **0.2656**     | **0.5928**        |
>
> [1]  [A Cheaper and Better Diffusion Language Model with Soft-Masked Noise](https://arxiv.org/abs/2304.04746)
>
> [2] [DiffusionBERT: Improving Generative Masked Language Models with Diffusion Models](https://arxiv.org/abs/2211.15029)
>
> [3] [Structured Denoising Diffusion Models in Discrete State-Spaces](https://arxiv.org/abs/2107.03006)
>
> [4] [Diffusion-LM Improves Controllable Text Generation](https://arxiv.org/abs/2205.14217)
>
> > Q3: If the first keywords are not right, the whole generation will become serious, how to avoid this phenomenon
>
> A3: Please note that we do not use a left-to-right autoregressive decoding strategy. As a non-autoregressive model, keywords are also generated step-by-step through sampling. If keywords are generated incorrectly, the model still has a certain probability of correcting the results in subsequent diffusion sampling steps. For example, in Table 3 of the paper, the model corrects "patrollingnine" to the proper keyword "geologist." On the other hand, if this method does not generate keywords well, the original approach of first generating irrelevant words may introduce more noise interference, making it even harder to generate the correct words.
>
> > Q4: What the advantages and disadvantages over the efficient methods, like dictionary-based methods or tf-idf-based methods.
>
> A4：Dictionary-based or tf-idf-based methods, and we even considered using PMI metrics, are just different standards for evaluating the information content of each word in a sentence. As stated in A2, our innovation lies more in discovering the gap and addressing it through the "keyinfo-first" strategy. Comparing this paper with information evaluation methods is unnecessary and unreasonable, as they represent work on different levels and in different directions. Additionally, we also tried using the tf-idf metric to measure word information in early experiments. Our method was still effective; the model could still exhibit a generation process prioritizing keywords, and generation quality could still be improved (8% improvement on the QQP dataset).
>
> In general, we extend our heartfelt gratitude to the reviewer for providing us with such insightful comments.  We will revise the paper according to your recommendations to make it more concise and convincing. We sincerely anticipate the possibility of an improved rating for the paper due to your valuable insights. Thank you again for your time and effort.

---

### Official Review · Reviewer_SEBp · 2023-08-05

**Soundness:** 3

**Excitement:**

3: Ambivalent: It has merits (e.g., it reports state-of-the-art results, the idea is nice), but there are key weaknesses (e.g., it describes incremental work), and it can significantly benefit from another round of revision. However, I won't object to accepting it if my co-reviewers champion it.

**Missing References:**

[2] Diffusion-NAT: Self-Prompting Discrete Diffusion for Non-Autoregressive Text Generation

**Paper Topic And Main Contributions:**

This paper proposes an information entropy aware diffusion model for NAR text generation. It considers the keyinfo-first generation strategy like humans into the existing continuous diffusion models, and revises the schedule of the diffusion model. Experimental results have shown the effectiveness of the proposed approach, on not commonly-used text generation datasets.

**Questions For The Authors:**

Please refer to reasons to reject

**Reasons To Accept:**

1.the focused keyinfo-first generation manner is indeed an important habit in humans to represent the spoken language, and the proposed new schedule can alleviate this issue.

**Reasons To Reject:**

1.continuous diffusion is not an effective way to generate natural language, as it cannot capture the discrete semantic information within natural language. Besides, it also greatly increases the cost to generate the text, as it requires to sample for multiple times for denoising. Recent work[1][2] has shown that discrete diffusion may be a good choice, and can outperform continuous diffusion methods. Authors should try its approach on these methods and show the generality of it.

2.the novelty of this work is worth concerning, as the new schedule is similar to the used one in diffusionBERT[1]. Besides, using self-condition is not a novel way, as it has also been widely-used in diffusion models, e.g., Diffusion-NAT[2].

3.the experiments are conducted on some unpopular datasets, but not the commonly-used summarization, dialog and question generation, such as XSum, Persona-chat and Squad. Although DiffuSeq has used these datasets, readers would be more concerned about the performance of all these methods on the popular datasets. I suggest authors to follow the used datasets and experimental settings in Diffusion-NAT[2].


[1] DiffusionBERT: Improving Generative Masked Language Models with Diffusion Models

[2] Diffusion-NAT: Self-Prompting Discrete Diffusion for Non-Autoregressive Text Generation

**Reproducibility:**

3: Could reproduce the results with some difficulty. The settings of parameters are underspecified or subjectively determined; the training/evaluation data are not widely available.

**Reviewer Confidence:**

4: Quite sure. I tried to check the important points carefully. It's unlikely, though conceivable, that I missed something that should affect my ratings.

---

> ### Author Rebuttal · Authors · 2023-08-27
>
> Thank you for the thoughtful feedback on my paper submission. We appreciate you taking the time to provide constructive comments. We summarize your main concerns as follows:
>
> > Q1: Continuous diffusion is not an effective way and try the proposed approach on discrete diffusion model.
>
> A1: Regarding the use of continuous diffusion, you raise a fair point about discrete diffusion being potentially more effective for natural language generation. I agree it is an interesting area for exploration, and have already run experiments using discrete diffusion methods as you suggest. To apply the proposed method to discrete diffusion models, we adopted some settings from DiffusionBERT, and adjusted the noise schedule and decoding schemes. On one hand, we trained our model on LM1B which was used by DiffusionBERT, and the experimental results are shown in Table 1 and Table 2. On the other hand, we also compared the performance of DiffusionBERT on the Quasae-T and QQP datasets used in our paper, and the experimental results are shown in Table 3.
>
> Table 1 Experimental results on LM1B
>
> | Method                 | PPL$\downarrow$ | BLEU$\uparrow$ | Self-BLEU$\downarrow$ |
> | ---------------------- | --------------- | -------------- | --------------------- |
> | D3PM[1]                | 77.50           | 0.4241         | 0.2288                |
> | Diffusion-LM[2]        | 118.62          | 0.3553         | 0.2668                |
> | DiffusionBERT          | **63.78**       | 0.4358         | 0.2151                |
> | InfoDiffusion-discrete | 68.46           | **0.4529**     | **0.2084**            |
>
> Table 2 Decoding process on LM1B
>
> | Method                 | Decoding Process                                             |
> | ---------------------- | ------------------------------------------------------------ |
> | DiffusionBERT          | [MASK] [MASK] [MASK] [MASK] [MASK] [MASK] [MASK] [MASK] [MASK] [MASK] |
> |                        | [MASK] , [MASK] [MASK] [MASK] [MASK] [MASK] that [MASK] .    |
> |                        | today , [MASK] will be [MASK] [MASK] that [MASK] .           |
> |                        | today , [MASK] will be remembered for that mistake .         |
> |                        | today , he will be remembered for that mistake .             |
> | InfoDiffusion-discrete | [MASK] [MASK] [MASK] [MASK] [MASK] [MASK] [MASK] [MASK]      |
> |                        | I [MASK] [MASK] [MASK] [MASK] [MASK] mistakes [MASK]         |
> |                        | I [MASK] [MASK] [MASK] their [MASK] mistakes .               |
> |                        | I make up for their [MASK] mistakes .                        |
> |                        | I make up for their recent mistakes .                        |
>
> Table3 Experimental results on Quasar-T and QQP dataset
>
> | Dataset  | Method        | BLEU$\uparrow$ | Rouge-L$\uparrow$ |
> | -------- | ------------- | -------------- | ----------------- |
> | Quasar-T | DiffuSeq      | 0.1731         | 0.3665            |
> |          | DiffusionBERT | 0.0971         | 0.3420            |
> |          | InfoDiffusion | **0.1924**     | **0.3892**        |
> | QQP      | DiffuSeq      | 0.2431         | 0.5880            |
> |          | DiffusionBERT | 0.2420         | 0.5845            |
> |          | InfoDiffusion | **0.2656**     | **0.5928**        |
>
> The experimental results show that in the discrete domain, the method we proposed is also effective. The model can follow the "keyinfo-first" generation order, and the generation quality is high. This demonstrates that our method has general applicability.
>
> [1] [Structured Denoising Diffusion Models in Discrete State-Spaces](https://arxiv.org/abs/2107.03006)
>
> [2] [Diffusion-LM Improves Controllable Text Generation](https://arxiv.org/abs/2205.14217)
>
> > Q2: The novelty of this work.
>
> A2: Concerning the novelty of this work, I understand your concern about similarities to prior work like Diffusion-BERT. However, our innovation lies more in the following: We discovered the phenomenon that diffusion models tend to generate meaningless words first while humans tend to think of key words first, and proposed the "keyinfo-first" generation strategy to prioritize generating key words first to address this issue. To measure whether a word is a key word, we hypothesize that words with higher information entropy are more critical and important to a sentence. Therefore, we use information entropy to control the priority. Our noise schedule is indeed similar to DiffusionBERT, but DiffusionBERT only utilizes information entropy to design noise while we explore how to guide diffusion models to generate in a more reasonable, effective and interpretable way. Following the "keyinfo-first" idea but using different computational formulas is also feasible. We also considered using tf-idf or PMI for the computation.
>
> Regarding "self-conditioning", we are not the first to adopt this technique. The earliest work introducing it to text generation can be traced back to [1]. However, as stated in lines 288-293 of the paper, we use this method because: in experiments we find that incorporating self-conditioning leads to more unified semantics and higher relevance between the texts generated at adjacent time steps. We are the first to discover this phenomenon and thereby utilize this technique accordingly. On the other hand, we not only tried self-conditioning, but also experimented with methods like anchor loss, noise factor, embedding normalization [2], etc. From the experimental results in the table below, self-conditioning achieves the best performance and is more interpretable, so we finally retained this technique.
>
> | Model                          | BLEU$\uparrow$ |
> | ------------------------------ | -------------- |
> | InfoDiffusion + self-condition | 0.3941         |
> | + anchor loss                  | 0.3742         |
> | + noise factor                 | 0.3613         |
> | + embedding normalization      | 0.3602         |
>
> [1] [Self-conditioned Embedding Diffusion for Text Generation](https://arxiv.org/abs/2211.04236)
>
> [2] [Difformer: Empowering Diffusion Models on the Embedding Space for Text Generation](https://arxiv.org/abs/2212.09412)
>
> > Q3: Follow the used datasets and experimental settings in Diffusion-NAT.
>
> A3: For the experiments, you make a good recommendation to evaluate on more standard datasets like XSum, Persona-Chat and SQuAD. The reason we did not use those datasets is that our work was completed around March, when most text diffusion models at that time were trained on the datasets we chose (e.g. DiffuSeq, Seqdiffuseq[1]), also for convenient comparison with concurrent works. On the other hand, Diffusion-NAT came out in May, after our work. Moreover, works concurrent with and even after Diffusion-NAT also used the same datasets as ours, such as RenderDiffusion[2], DiffuSIA[3], TESS[4] etc. We are now training and evaluating our model on the datasets you mentioned as quickly as possible, but due to limited GPU resources etc., we do not yet have final results.
>
> [1] [SeqDiffuSeq: Text Diffusion with Encoder-Decoder Transformers](https://arxiv.org/abs/2212.10325)https://arxiv.org/abs/2305.04044)
>
> [2] [GlyphDiffusion: Text Generation as Image Generation](https://arxiv.org/abs/2304.12519)
>
> [3] [DiffuSIA: A Spiral Interaction Architecture for Encoder-Decoder Text Diffusion](https://arxiv.org/abs/2305.11517)
>
> [4] [TESS: Text-to-Text Self-Conditioned Simplex Diffusion](https://arxiv.org/abs/2305.08379)
>
> Thank you again for the helpful feedback. We sincerely hope the changes outlined in our rebuttal and planned for the revision will adequately address your concerns and result in an improved rating. Your thoughtful suggestions substantially enhance the quality of our work. We greatly appreciate reviewers like yourself who uphold rigorous standards and provide guidance to advance research. Please let us know if you have any other feedback on how we can further improve the paper.

---

### Official Review · Reviewer_xSvW · 2023-08-09

**Soundness:** 3

**Excitement:**

4: Strong: This paper deepens the understanding of some phenomenon or lowers the barriers to an existing research direction.

**Missing References:**

Diffused Conditional Sequence Learning by Manipulating Noises, Ye et al., 2023
Seqdiffuseq: Text diffusion with encoder-decoder transformers, Yuan et al., 2022
A cheaper and better diffusion language model with soft-masked noise, Chen et al., 2023

**Paper Topic And Main Contributions:**

This paper introduces InfoDiffusion, a non-autoregressive text diffusion model that aims to bridge the disparities between the text generation process seen in diffusion models and the innate text generation process of humans. InfoDiffusion presents a novel "key info-first" generation strategy and integrates a noise schedule that adapts based on the volume of textual information. This is achieved through the fusion of self-conditioning with a recently introduced partially noising model framework. The empirical findings illustrate enhancements over the baseline model in aspects of generation quality, diversity, and the efficiency of sampling.

**Reasons To Accept:**

1. A new trial to fill the gaps between text generation process of diffusion models with natural text generation process of humans.
2. Introducing a “key info-first” generation strategy and incorporates a noise schedule based on the amount of text info. It combines self-conditioning with a newly proposed partially noising model structure.
3. Experimental results show the improvements over the baseline model in terms of generation quality, diversity and sampling efficiency.

**Reasons To Reject:**

1. The absence of an extensive incorporation of relevant studies concerning the exploration of decoding order in diffusion models, such as SeqDiffuSeq (Seqdiffuseq: Text diffusion with encoder-decoder transformers), DiNoiSer (Diffused Conditional Sequence Learning by Manipulating Noises), and Masked-Diffuse LM (A cheaper and better diffusion language model with soft-masked noise), which have similarly investigated alternative corruption and decoding processes.
2. Inadequate comparison with other closely related methodologies that delve into the decoding order of diffusion models, coupled with a limited depth of analysis regarding distinctive aspects and innovative elements in comparison to other similar systems.

Diffused Conditional Sequence Learning by Manipulating Noises, Ye et al., 2023
Seqdiffuseq: Text diffusion with encoder-decoder transformers, Yuan et al., 2022
A cheaper and better diffusion language model with soft-masked noise, Chen et al., 2023

**Reproducibility:**

4: Could mostly reproduce the results, but there may be some variation because of sample variance or minor variations in their interpretation of the protocol or method.

**Reviewer Confidence:**

4: Quite sure. I tried to check the important points carefully. It's unlikely, though conceivable, that I missed something that should affect my ratings.

---

> ### Author Rebuttal · Authors · 2023-08-27
>
> Thank you for taking the time to provide thoughtful feedback on our paper submission. We greatly appreciate you highlighting the strengths of our work, including the novelty of our approach for bridging the gap between diffusion models and human text generation, as well as the improvements shown over baseline models.
>
> Regarding your comments about missing incorporation of relevant studies. We will add a chapter in the appendix. For examlpe: "SeqDiffuSeq[1] proposes an encoder-decoder diffusion model architecture for conditional generation and uses adaptive noise schedule technique to improve generation quality.  DiNoiSer[2] proposes an adaptive method to determine the range of noise scales sampled for counter-discreteness training allowing the model to leverage amplified noise scales from the source conditions during inference. Masked Diffuse LM[3] follows easy first generation and designs a soft masking strategy based on tf-idf. Diffusion-NAT[4] proposes a iterative self-prompting strategy for denoising process. GlyphDiffusion[5] render a target text onto a glyph image containing visual language content and generate an image containing the target text through diffusion decoding process."
>
> Regarding the "distinctive aspects and innovative elements", most prior works on designing different decoding techniques focus on sentence-level structure and process. For example, combining encoder-decoder structures or adjusting the overall noise level of the entire sentence. In contrast, we pay more attention to the decoding order at the word-level. We discover the phenomenon that diffusion models tend to generate meaningless words first, which differs from humans' language habit of thinking about key words first. Therefore, we propose the "keyinfo-first" decoding strategy, which is more reasonable and interpretable. On the other hand, our proposed keyinfo-first strategy can be combined with sentence-level decoding techniques to achieve better decoding performance. We are also actively conducting experiments to validate this point, such as adopting the "keyinfo-first" decoding strategy on top of the encoder-decoder model structure proposed in SeqDiffuSeq, and verifying the general applicability of our proposed strategy on discrete diffusion models.
>
> Overall, we found your feedback extremely constructive. Please let us know if you have any other suggestions for improving the completeness and quality of our submission. We appreciate you taking the time to provide such thoughtful and detailed comments.
>
> [1] [SeqDiffuSeq: Text Diffusion with Encoder-Decoder Transformers](https://arxiv.org/abs/2212.10325)
>
> [2] [DINOISER: Diffused Conditional Sequence Learning by Manipulating Noises](https://arxiv.org/abs/2302.10025)
>
> [3] [A Cheaper and Better Diffusion Language Model with Soft-Masked Noise](https://arxiv.org/abs/2304.04746)
>
> [4] [Diffusion-NAT: Self-Prompting Discrete Diffusion for Non-Autoregressive Text Generation](https://arxiv.org/abs/2305.04044)
>
> [5] [GlyphDiffusion: Text Generation as Image Generation](https://arxiv.org/abs/2304.12519)

---

### Meta-Review · Area_Chair_osNS · 2023-09-17

**Recommendation:** 4

**Metareview:**

Summary:

The paper presents a novel “key-info-first” continuous diffusion model and an information-entropy aware noise schedule for improving open-ended non-autoregressive text generation. The paper is generally well written, and the proposed approach seems sound. Experiments show positive results against selected baselines using automatic metrics like perplexity, BLEU, ROUGE, and BERTScore. No human evaluations were conducted to boost the claims. Additional experiments with discrete diffusions and other datasets were provided during rebuttal period, showing potential for an improved revised paper.

Strengths:

1. Proposing a new diffusion process that is sensible for text generation by prioritizing keywords first, showing improved performance through automatic metrics.
2. Proposing an information-entropy aware noise schedule
3. Experiments with automatic metrics, in addition to some qualitative examples, demonstrating the effectiveness of the proposal compared to selected baselines.


Weaknesses:

1. The overall presentation and writing should be improved based on reviewers’ comments.
2. The paper needs to incorporate the additional experiments and analyses provided during the rebuttal period in the final revision.
3. Missing human evaluation

---

### Decision · Program_Chairs · 2023-10-07

**Decision:**

Accept-Findings

**Comment:**

Summary:

The paper presents a novel “key-info-first” continuous diffusion model and an information-entropy aware noise schedule for improving open-ended non-autoregressive text generation. The paper is generally well written, and the proposed approach seems sound. Experiments show positive results against selected baselines using automatic metrics like perplexity, BLEU, ROUGE, and BERTScore. No human evaluations were conducted to boost the claims. Additional experiments with discrete diffusions and other datasets were provided during rebuttal period, showing potential for an improved revised paper.

Strengths:

1. Proposing a new diffusion process that is sensible for text generation by prioritizing keywords first, showing improved performance through automatic metrics.
2. Proposing an information-entropy aware noise schedule
3. Experiments with automatic metrics, in addition to some qualitative examples, demonstrating the effectiveness of the proposal compared to selected baselines.


Weaknesses:

1. The overall presentation and writing should be improved based on reviewers’ comments.
2. The paper needs to incorporate the additional experiments and analyses provided during the rebuttal period in the final revision.
3. Missing human evaluation